# SW-GAN: Road Extraction from Remote Sensing Imagery Using Semi-Weakly Supervised Adversarial Learning

**Hao Chen** **, Shuang Peng, Chun Du \*, Jun Li and Songbing Wu**

College of Electronic Science and Technology, National University of Defense Technology, No. 109 Deya Road, Changsha 410073, China; hchen@nudt.edu.cn (H.C.); pengshuang08@nudt.edu.cn (S.P.); junli@nudt.edu.cn (J.L.); wusongbing10@nudt.edu.cn (S.W.)
**\*** Correspondence: duchun@nudt.edu.cn

**Abstract:** Road networks play a fundamental role in our daily life. It is of importance to extract the road structure in a timely and precise manner with the rapid evolution of urban road structure. Recently, road network extraction using deep learning has become an effective and popular method. The main shortcoming of the road extraction using deep learning methods lies in the fact that there is a need for a large amount of training datasets. Additionally, the datasets need to be elaborately annotated, which is usually labor-intensive and time-consuming; thus, lots of weak annotations (such as the centerline from OpenStreetMap) have accumulated over the past a few decades. To make full use of the weak annotations, we propose a novel semi-weakly supervised method based on adversarial learning to extract road networks from remote sensing imagery. Our method uses a small set of pixel-wise annotated data and a large amount of weakly annotated data for training. The experimental results show that the proposed approach can achieve a maintained performance compared with the methods that use a large number of full pixel-wise annotations while using less fully annotated data.

**Keywords:** road extraction; semi-weakly supervised learning; Generative Adversarial Network; OpenStreetMap

## 1. Introduction

Accurate and up-to-date road network information is fundamental for our daily life, whether for use in urban management, traffic planning, vehicle navigation, intelligent transportation systems, and so on. Therefore, continually extracting road networks is of great importance, especially for fast-growing areas. With the rapid development of remote sensing technology, images can be easily obtained from remote sensors installed on drones or satellites, which allows the updating of road networks and the timely adoption of road extraction methods of remote sensing images.

Recently, deep learning methods have been leading in road extraction methods for remote sensing images [1,2]. However, the superior performance of these deep learning-based methods highly depends on massive quantities of precisely annotated training data [3]. Obtaining a large amount of precisely annotating data is usually a labor-intensive and time-consuming process. Thus, it is hard to acquire large-scale precisely annotated data in reality, especially since weak annotations (such as road centerline) can be easily and economically obtained, given the rapid development of the OpenStreetMap (OSM).

Remote sensing images with different types of annotations are shown in Figure 1. Figure 1a represents a typical image obtained from remote sensing cameras; Figure 1b represents the full labelled image which is annotated by a human pixel-by-pixel; and Figure 1c represents the road centerline, which can be easily obtained form OSM [4–6] and can be regarded as sparse scribbles for road labels.

From Figure 1, we can see that the precise labelled annotation is much more complicated than the scribble OSM centerline.

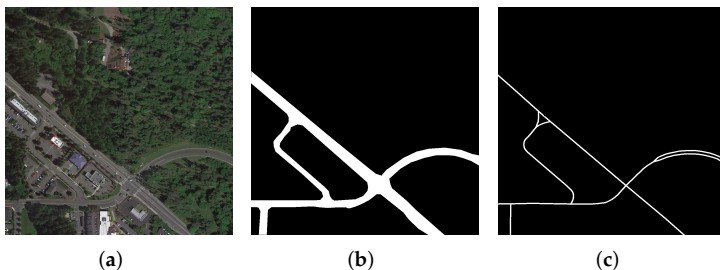

**Figure 1.** Sample images. (**a**) is the satellite image, (**b**) is pixel-wise annotation and (**c**) is OSM centerline which can be regarded as a sparse annotation.

To mitigate the weakness of the massive demand for high-quality annotated training data in deep learning methods, Wu et al. and Wei et al. only use the weak annotations from OSM centerline to extract the road networks [4,6]. Nevertheless, there is still room for improvement in the performance of these methods, since they do not use any existing data with full annotations.

In order to reduce the reliance on high-quality pixel-wise annotations and to improve road extraction performance for automated satellite images interpretation, this paper proposes a semi-supervised method named Semi-Weakly Generative Adversarial Network (SW-GAN) to extract the road network from remote sensing imagery. In this method, to make efficient road extraction, we use only a small amount of precisely annotated data and large amounts of easily acquired weakly annotated data (OSM centerlines).

The contributions of this paper are summarized as:

1. A novel semi-weakly supervised learning framework is proposed for extracting the road networks, which can make good use of both the massive amounts of weakly labelled annotations and small amounts of precisely annotated data.

2. To make good use of the weak annotations, we design a novel weakly supervised network and add it to the typical GAN (Generative Adversarial Network) network. The weakly supervised network can automatically learn from the latent features of a large amount of weakly supervised data and feed the learned features into the GAN networks.

3. To validate the proposed method, we carry out extensive experiments on three real-world datasets. The results show that the proposed semi-weakly supervised method can obtain very close results to that of the fully supervised methods while just using 20% fully annotated data.

The following sections are arranged as follows. First, the related work of road extraction, semi-supervised and weakly supervised learning are discussed. Then, the proposed method, SW-GAN, is illustrated. After that, the experiments and discussions are presented. Finally, the conclusion of this paper is drawn.

## 2. Related Work

### 2.1. Road Extraction

Traditionally, road extraction from remote sensing images can be divided into unsupervised and supervised methods.

The main difference between the unsupervised road extraction methods and supervised methods lies in the fact that the unsupervised ones do not need annotated data for training. Hence, the unsupervised methods mainly use clustering technologies to extract the road network. K-means [7], graph theory [8,9], mean shift [10,11] and domain mapping [12,13] are widely used as unsupervised learning algorithms to extract the road networks. However, the performance of unsupervised methods in road extraction remains lower than the supervised ones.

Unlike unsupervised road extraction methods, supervised approaches usually need lots of training samples. The extraction performance heavily relies on the quality and amount of the training samples. These methods mainly include support vector machine

(SVM) [14], conventional artificial neural network (ANN) [15,16], Markov random field (MRF) [17] and Bayesian decision theory [18].

In the past few decades, Deep Learning (DL) methods have promoted the development of many image processing problems such as image semantic segmentation, change detection, target tracking and so on. Additionally, the deep convolutional neural networks (DCNN) [19,20] are typical DL methods used in the image semantic segmentation. Recently, many DL methods have made great progress in road extraction [1,2]. A dual-generation GAN(DH-GAN) network was proposed to extract the road topologies by D. Costea et al. [21]. The DH-GAN has significantly improved the topology and accuracy of the road networks by using SBO (Smoothing-Based Optimization) methods . YY.Xu et al. proposed a road extraction method by adding both the global and local attention unit into U-Net and DenseNet [22] (GL-Dense-U-Net) [23]. Z. Zhang et al. proposed a method by integrating the strengths of residual learning and U-Net [14]. Q. Zhu et al. designed a Global Context-aware and Batch-independent Network (GCB-Net) to extract complete and continuous road networks from remote sensing images [24]. The GCB-Net can eliminate the batch dependency to accelerate learning and further improve the robustness of the model. Y.X. Xu et al. proposed a spatial attention-based road extraction network and employed signed distance between roads and buildings to enhance the extraction accuracy for the potential roads around the thorny occlusion areas of remote sensing images [25]. Z. Chen et al. modified the architecture of U-Net, and designed an asymmetric encoder-decoder network for road extraction from remote sensing images [26]. J. Zhang et al. employed a DenseNet [22]-based network with a data fusion mechanism to conduct low-grade road extraction from optical and SAR images [27].

Furthermore, the development of road extraction has been promoted rapidly through competition [28]. For instance, many methods have effectively promoted the performance of road extraction [29,30]. L. Zhou et al. won a competition by using a neural network named D-LinkNet [31], which used an encoder–decoder structure architecture with dilated convolution.

Although the performance of road extraction has been greatly improved by using these DL-based methods, there are still limitations owing to the demand for a massive amount of precisely annotated training data. It is still a labor-intensive and time-consuming process to generate plentiful high-quality training data, as the performance of DL-based road extraction methods relies heavily on a large amount of high-quality labelled training data.

### 2.2. Semi-Supervised and Weakly Supervised Deep Learning Methods

In order to mitigate the dependency on pixel-wise annotation training data for deep learning-based models, some semi-supervised learning and weakly supervised methods have been proposed. These models can promote segmentation efficiency instead of using just fully precise annotated datasets as supervision.

The semi-supervised model learned from limited pixel-wise annotated samples and utilized huge unlabeled data to improve the performance of remote sensing image segmentation. The most popular semi-supervised models in the remote sensing image semantic segmentation task are GAN-based and pseudo-labels-based methods. W. Hung et al. proposed a GAN-based network that can learn from unlabelled samples [32]. S. Mittal et al. proposed a dual-branch semi-supervised approach for semantic segmentation [33], which can reduce both the low-level and the high-level artifacts when training with a few labels. J.X. Wang et al. presented an iterative contrastive network for remote sensing image semantic segmentation, which can continuously learn more potential information from labeled samples and generate better pseudo-labels for unlabeled data [34]. S. Desai et al. employed active learning techniques to generate pseudo-labels from a small set of labeled examples which are used to augment the labeled training set, and enhanced the performance of remote sensing semantic segmentation network [35].

Compared to the pixel-wise annotations, weak annotations (such as scribbles [4,36,37], bounding boxes [38], points [39,40] and image-level tags [41,42]) were much easier to obtain.

Therefore, weakly supervised learning was more popular in segmentation tasks [2,43]. S. Wu et al proposed a novel model named MD-ResUnet, which used only OSM centerline as weak annotations and achieved good performance in road extraction from remote sensing images [4]. J. Zhang et al. adopted crowd-sourced GPS trajectory data as weak annotations to promote the road extraction performance from aerial imagery [44].

These semi-supervised or weakly supervised road extraction methods have truly reduced the demand for a large number of precisely annotated data. However, there is still a gap in road extraction performance between these methods and fully supervised methods.

In fact, for semi-supervised models, the unlabeled data can be replaced by weakly annotated data, and this type of method is called semi-weakly supervised learning [45]. Compared to conventional semi-supervised models, semi-weakly supervised models are supposed to obtain better performance, since the weakly annotated data has more potential information than that of the unlabeled data [46].

To the best of our knowledge, there is still no appropriate semi-weakly supervised learning method can be used to promote the extraction performance by combining a few full pixel-wise annotations and lots of weak annotations such as scribble annotations in the remote sensing image road extraction task. We want to design a semi-weakly supervised learning model for road extraction, and try to make its performance close to that of fully supervised models.

## 3. Methodology for SW-GAN Networks Using in Road Extraction

In this paper, a semi-weakly supervised method named SW-GAN is proposed to improve the performance of road extraction. For SW-GAN, the input of the training dataset can be divided into two parts. One is satellite images with high-quality pixel-wise annotations, which is the fully supervised dataset. The other is satellite images with weakly scribbled supervision (OSM centerline), the weakly supervised dataset. Usually, the number of samples of the fully supervised dataset is much smaller than that of weakly supervised dataset.

The overall view of the SW-GAN method is shown in Figure 2. Unlike the standard architecture of GAN, SW-GAN consists of three main components: a fully supervised generator, a weakly supervised generator, and a discriminator. The fully supervised generator is fed into both the fully supervised dataset and weakly supervised dataset to learn the road features and produce a realistic road network. The weakly supervised generator is only fed with the weakly supervised dataset to learn the potential distribution of road network. The weakly supervised generator can feed the learned features into the fully supervised generator. The discriminator is used to distinguish whether the produced road network is a pixel-wise human-annotated road network or whether it was synthesized by the fully supervised generator. Hence, the fully supervised generator and discriminator are pitted against each other as adversaries.

As Figure 2 shows, the weakly supervised dataset is fed into both the fully supervised and weakly supervised generators simultaneously. We try to make the output of the fully supervised generator and weakly supervised generator similar to each other. This strategy ensures the consistency of the output of the fully supervised generator and the weakly supervised generator, and makes effort to feed the learned features from the weakly supervised generator into the fully supervised generator. The fully supervised dataset is only fed into the fully supervised generator. The training process is a standard supervised manner.

This mechanism guarantees that the fully supervised generator can learn features from both the fully supervised dataset and weakly supervised dataset. Therefore, it can extract more potential details from the road and prevent the algorithm from over-fitting.

The details of the components of SW-GAN and how to make the weakly supervised dataset are illustrated in the following subsections.

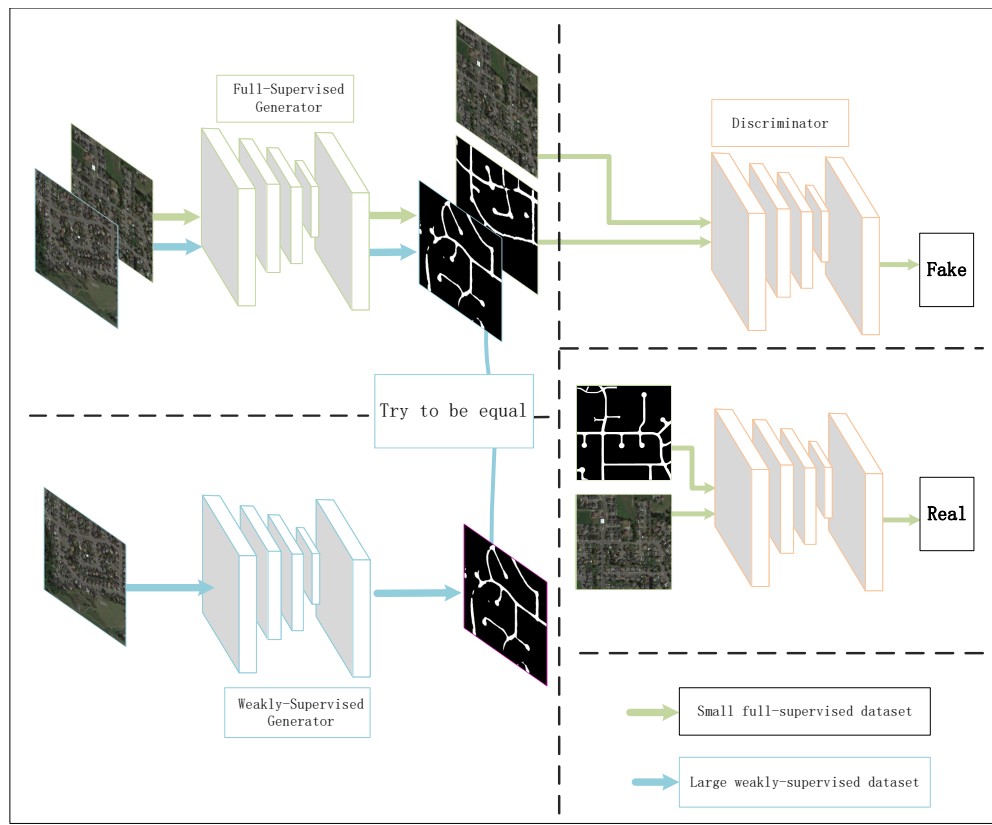

**Figure 2.** Overall framework of the proposed approach.

### 3.1. The Weak Road Annotation Inference from OSM Data

Although OSM data is a widely used crowdsource open data set, it still includes some incorrect or incomplete road centerlines (not always in the center of the real road). Therefore, we cannot obtain accurate weak annotations using only the OSM road centerline. We need to do some preprocessing for OSM data to form weak annotated data for road extraction. Firstly, we project the remote sensing images and corresponding OSM road centerline data to the same geographic coordinates system to keep them geographically consistent. Secondly, the initial road annotations are inferred from the centerline using the prior knowledge of the image resolution and the grade of a road. Since different grades of roads usually correspond to different road widths, the upper bound of road width can be inferred from road grades.

As shown in Figure 3, the road and non-road are inferred by the grade and centerline of a road. Additionally, the other pixels that cannot be determined by the distance to the road centerline are labeled as unknown. In this paper, we set the pixels within 7 m of the road centerline as road and the pixels beyond 50 m as nonroad.

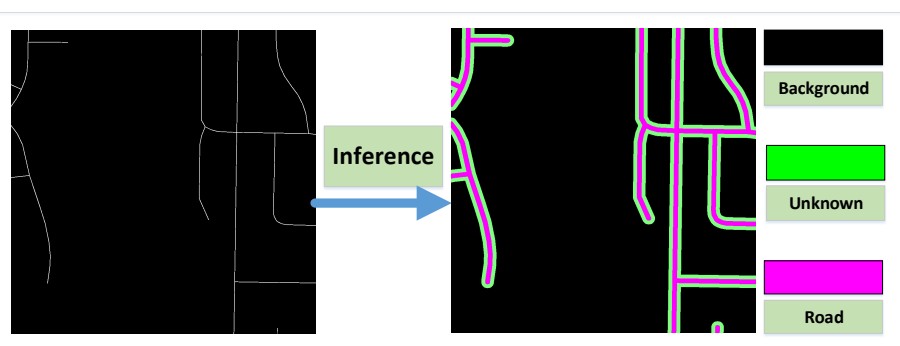

**Figure 3.** Weak road annotation inference.

### *3.2. The Components of SW-GAN*

We denote the fully supervised generator as $G_f$, the weakly supervised generator as $G_w$, the discriminator as $D$, the images with weak annotation as $X_w$ and the images with full annotation as $X_f$. The corresponding annotations of $X_w$ and $X_f$ are $Y_w$ and $Y_f$, respectively.

Like most GAN models, the discriminator is used to distinguish the composite data and real data, while the fully supervised generator is used to generate road networks and deceive the discriminator. The fully supervised generator and the discriminator form a conditional GAN model. The objective of the conditional GAN in this paper can be described as Equation (1):

$$\mathcal{L}_{cGAN}(G_f, D) = E_{X_f, Y_f}[log D(X_f, Y_f)] + E_{X_f}[log(1 - D(X_f, G_f(X_f)))]. \tag{1}$$

From Equation (1), we can see that the fully supervised generator and the discriminator play a two-player mini-max game. Equation (1) consists of two parts: the first term is the loss function for discriminator, and the second term is the loss function for the fully supervised generator.

#### 3.2.1. The Fully Supervised Generator

The fully supervised generator uses the D-linknet [31] as backbone in this paper, which adopts an encoder–decoder structure, dilated convolution and pre-trained encoder for the road extraction. Furthermore, this has been used in road extraction works [31,47]. As the fully supervised generator of SW-GAN is tasked with approaching the ground truth of fully supervised dataset $X_f$ ($Y_f$), we employ cross entropy to measure the fully supervised loss $\mathcal{L}_f$ as Equation (2):

$$\mathcal{L}_f(G_f) = - \sum_{x_f^{(i)} \in X_f} y_f^{(i)} log(G_f(x_f^{(i)})), \tag{2}$$

where $y_f^{(i)}$ is the corresponding label of $x_f^{(i)}$.

#### 3.2.2. The Weakly Supervised Generator

To make full use of the weak annotations, the weakly supervised generator refers to the method used in weakly supervised road extraction [4] and weakly supervised semantic segmentation [48]. These methods use the high-order information and the relationship between pixels of the image. The weakly supervised generator of SW-GAN is shown in Figure 4.

The weakly supervised method improves the road extraction performance by using the normalized cut loss to reflect the high-order information [4]. Additionally, we add a learnable pyramid dilation network to weakly supervised generator to expand the receptive field of convolution of conventional ResUnet. The weakly supervised generator is supervised by the road weak annotations. We adopt the loss function in [4] for the weakly supervised generator, and noted it as $L_w(G_w)$ in this article.

Additionally, the fully supervised generator of the network is required to output the same results as that of the weakly supervised generator. We design a loss called consistency loss, which can be defined as Equation (3):

$$\mathcal{L}_{cnst}(G_f, G_w) = - \sum_{x_w^{(i)} \in X_w} G_w(x_w^{(i)}) log(G_f(x_w^{(i)})). \tag{3}$$

Equation (3) ensures the consistency between the fully supervised generator $G_f$ and the weakly supervised generator $G_w$ on the weakly supervised dataset $X_w$.

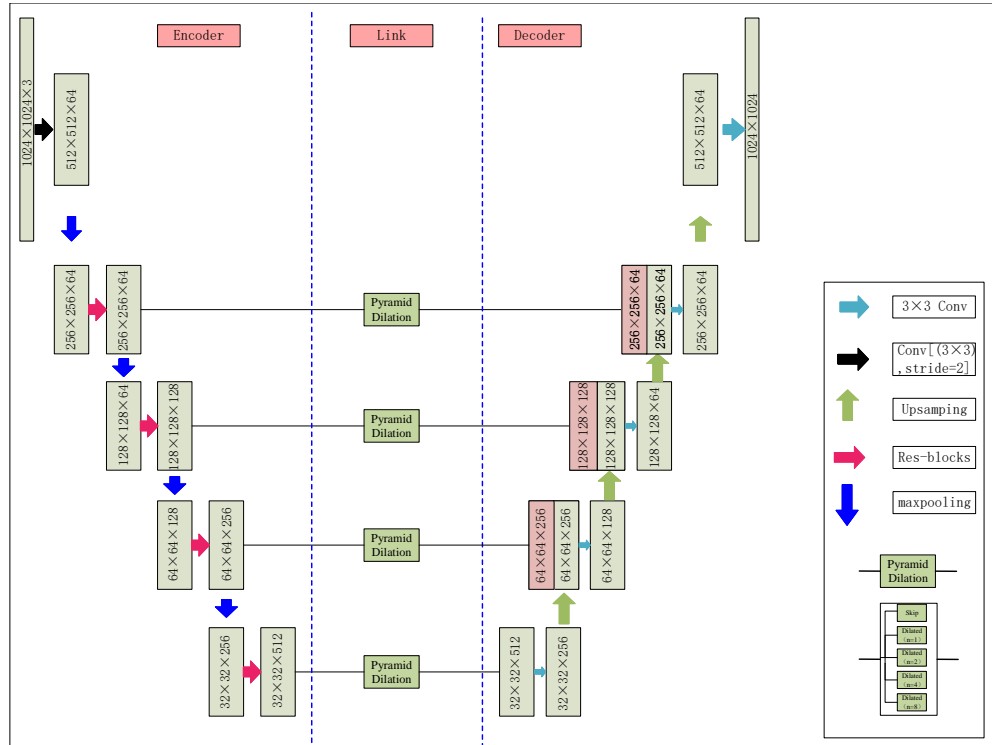

**Figure 4.** The architecture of the weakly supervised generator.

### 3.2.3. The Discriminator

Unlike the most discriminators of typical GANs, we adopt a fully convolutional network as our discriminator, since it can take inputs of arbitrary sizes. The input of the discriminator is the original satellite images and the road segmentation maps generated by the fully supervised generator (or the ground-truth road network labels). The output is a probability map of size H × W × 1.The pixel of the output probability map represents the possibility to be a real label. To train the discriminator, we minimize the loss $\mathcal{L}_d(D)$ as Equation (4):

$$\mathcal{L}_d(D) = -\sum_{x_f^{(i)} \in X_f} (1 - y_n)log(1 - D(G_f(x_f^{(i)}), x_f^{(i)})) + y_n log(D(y_f^{(i)}), x_f^{(i)}), \qquad (4)$$

where $y_n = 0$ if the sample is from the fully supervised generator network; $y_n = 1$ if the sample is from the ground truth label. $y_f^{(i)}$ is the corresponding label of $x_f^{(i)}$. Equation (4) is derived from Equation (1).

### 3.3. Loss Function and the Training Algorithm for SW-GAN

The final objective of SW-GAN can be formulated as Equation (5):

$$G_f^* = arg \min_{G_f} \max_D \mathcal{L}_{cGAN}(G_f, D) + \lambda_1 \mathcal{L}_f(G_f) + \lambda_2 \mathcal{L}_{cnst}(G_f, G_w). \qquad (5)$$

To minimize the loss for training SW-GAN, the proposed algorithm is shown in Algorithm 1. To balance the generators and the discriminator of SW-GAN, $G_w$ and $G_f$ are pre-trained using only the $\mathcal{L}_w$ and $\mathcal{L}_f$ independently when the iteration is less than the generator learning iteration $It_{Gl}$. When the iteration is larger than $It_{Gl}$, the $G_w$, $G_f$ and $D$ are trained using $\mathcal{L}_w$, $\mathcal{L}_{cGAN} + \lambda_1 \mathcal{L}_f + \lambda_2 \mathcal{L}_{cnst}$ and $\mathcal{L}_d$, respectively. This pre-trained mechanism can be regarded as a warm up process, which can effectively prevent the model from collapsing during the progress of training.

---

**Algorithm 1:** Training SW-GAN

---

   **input**　:

        Input datasets for $X_w$; $X_f$; $Y_w$; $Y_f$;

        The weight parameter of $\lambda_1$; $\lambda_2$;

        The learning rate $\alpha$;

        warm up times of training iteration $It_{Gl}$;

        the maximum times of training iteration $It_{max}$ ;

   **output**:

        The model parameters of the $\omega_{G_f}$, $\omega_{G_w}$, $\omega_D$ ;

1　randomly initialize the model parameter $\omega_{G_1}$, $\omega_{G_2}$, $\omega_D$;

2　$iternum = 0$

3　for $iternum < It_{max}$:

4　　　$if \quad iternum < It_{Gl}$ :

5　　　　　$\mathcal{L}(\omega_{G_w}) \leftarrow \mathcal{L}_w$

6　　　　　$\omega_{G_w} \leftarrow \omega_{G_w} - \alpha\nabla\mathcal{L}(\omega_{G_w})$

7　　　　　$\mathcal{L}(\omega_{G_f}) \leftarrow \mathcal{L}_f$

8　　　　　$\omega_{G_w} \leftarrow \omega_{G_f} - \alpha\nabla\mathcal{L}(\omega_{G_f})$

9　　　$else$ :

10　　　　　$\mathcal{L}(\omega_{G_w}) \leftarrow \mathcal{L}_w$

11　　　　　$\omega_{G_w} \leftarrow \omega_{G_w} - \alpha\nabla\mathcal{L}(\omega_{G_w})$

12　　　　　$\mathcal{L}(\omega_{G_f}) \leftarrow \mathcal{L}_{cGAN} + \lambda_1\mathcal{L}_f + \lambda_2\mathcal{L}_{cnst}$

13　　　　　$\omega_{G_w} \leftarrow \omega_{G_f} - \alpha\nabla\mathcal{L}(\omega_{G_f})$

14　　　　　$\mathcal{L}(\omega_D) \leftarrow \mathcal{L}_d$

15　　　　　$\omega_D \leftarrow \omega_D - \alpha\nabla\mathcal{L}(\omega_D)$

16　　　$iternum + = 1$

---

## 4. Experiment

### 4.1. Dataset Description

To prove the effectiveness of the proposed semi-supervised road extraction method, we conduct experiments on three separate datasets. The details for all these datasets are shown in Table 1.

Dataset 1 was collected from the real world in Seattle. There were 315 aerial images and corresponding centerline annotations and pixel-wise annotations in the dataset. The aerial images were collected from Google Earth, the centerline annotations were from the OSM, and we manually labelled the pixel-wise annotations pixel-by-pixel. This dataset covered both urban and rural regions. In the experiment, the whole dataset was divided into two parts. One part has 285 images, which are used as training dataset, and the remaining 30 images are used for testing. The resolution in the satellite images is 1.2 m. Most of satellite images are covered by a complex terrestrial environment such as occlusions, trees or shadows of buildings. Therefore, it is difficult to distinguish the background and roads.

The images and the full annotations of dataset 2 were released by Tao Sun et al. [47]. They manually generated the pixel-wise labels by labelling road pixels in the images. As the geographical coordinate of the images was given, the centerlines of roads were collected from OSM. There were 350 images, and the resolution of these images was 0.5 m.

Dataset 3 was released in the competition in the CVPR 2018 deep globe challenge [28]. There are 2048 training images and 300 testing images. The resolution of these images was 1 m. The precise annotations were manual labelled. As the geographical coordinates were

unknown, the centerlines of roads were generated by thinning the precisely annotated annotations.

As the DL methods require a large amount of training data, and the size of dataset 1 and dataset 2 is small, we generate synthetic datasets by altering original ones using diagonal flipping, horizontal flipping, scaling and image shifting. The training dataset is 6~8 times larger than the original one after the augmentation. This can prevent the training processing from over-fitting.

**Table 1.** Dataset Description (For dataset 1 and dataset 2, the numbers in parentheses are the amount of data after augmentation).

| Dataset | Resolution | Area | Train | Test | Image Origin | Full Annotation | Centerline |
|---------|-----------|------|-------|------|--------------|-----------------|------------|
| dataset 1 | 1.2 m | Seattle | 285 (1995) | 30 (155) | Google Earth | manual | OSM |
| dataset 2 | 0.5 m | Beijing | 350 (2100) | 27 (135) | Gaode Map | manual | OSM |
| dataset 3 | 1 m | America | 2048 | 300 | Google Earth | manual | image |

### 4.2. Baselines and Experiment Setting

Our proposed SW-GAN is implemented using Pytorch [49]. We train the model using 2 GTX1080Ti GPUs.

In this paper, ResUNet [14], D-Linknet [31], DeeplabV3 [50] and weakly supervised road extraction algorithm MD-ResUnet [4] are selected as our baseline. ResUNet [14] was built with residual units and had similar architecture to that of U-Net; D-linknet [31] was leading in CVPR2018 Digital Road Extraction Challenge which was proved to be efficient in road extraction. DeeplabV3 [50] is a very widely used network for image semantic segmentation, and has a good performance in the road extraction task from remote sensing images. MD-ResUnet [4] used only weak OSM centerline as annotation, which added the normalized cut loss to reflect the high-order information and a multi-dilation network to conventional ResUnet [14] to expand the receptive field of convolution.

All experiments are evaluated based on IoU (Intersection over Union) and mAP (mean Average Precision). IoU indicates the overlap of the predicted bounding box coordinates to the ground truth box, and it can be formulated as Equation (6)

$$IoU = \frac{TP}{TP + FP + FN}, \tag{6}$$

where $TP$, $FP$, $TN$ and $FN$ are the number of true positives, false positives, true negatives and false negatives, respectively. mAP is a metric used to evaluate object detection or semantic segmentation models. The definition of mAP is depicted in reference [51].

We use SGD [52] as the optimizer to train the proposed network, and initially set the learning rate as $2 \times 10^{-4}$. The batch size is set to 2, as this is the number of GPUs we used. Additionally, according to the experiment in Section 4.6, we set the parameter $\lambda_1$ as 0.1 and the cor-training hyper-parameter $\lambda_2$ as 2.

To prove the effectiveness of the proposed method, firstly, an experiment is conducted to evaluate the performance of SW-GAN. Then, we conduct an experiment to evaluate the Semi-Weak methods using different ratios of full training images. Finally, we employ an ablation study to validate the rationality of the design for SW-GAN.

### 4.3. Evaluation the Performance of SW-GAN

To validate the proposed SW-GAN, we compare the semi-supervised method SW-GAN with the weakly supervised model MD-ResUnet and fully supervised methods ResUnet, D-linknet and DeeplabV3. The training datasets include 20% full pixel-wise annotation samples and 80% weakly supervised annotation samples to train all the different models. The weakly supervised method MD-ResUnet was trained with the 80% weak annotation

samples, and the full supervised methods ResUnet and D-linknet were trained using the 20% full pixel-wise annotation samples. SW-GAN was trained using both the full supervised data and the weakly supervised data. The results are shown in Table 2 and Figures 5–7. In Figures 5–7, TN = true negative; FN = false negative; FP = false positive; TP = true positive.

**Table 2.** SW-GAN Performance Evaluation (bold: best).

| Model | | MD-ResUnet | ResUnet | D-Linknet | DeeplabV3 | SW-GAN |
|-------|-----------|------------|---------|-----------|-----------|--------|
| IoU | dataset 1 | 0.7387 | 0.7189 | 0.7293 | 0.7152 | **0.778** |
| | Dataset 2 | 0.5852 | 0.584 | 0.6028 | 0.5902 | **0.637** |
| | Dataset 3 | 0.656 | 0.6684 | 0.695 | 0.6585 | **0.714** |
| mAP | dataset 1 | 0.8264 | 0.8154 | 0.8562 | 0.8260 | **0.895** |
| | Dataset 2 | 0.7356 | 0.7405 | 0.7382 | 0.7154 | **0.738** |
| | Dataset 3 | 0.705 | 0.7851 | 0.799 | 0.7502 | **0.812** |

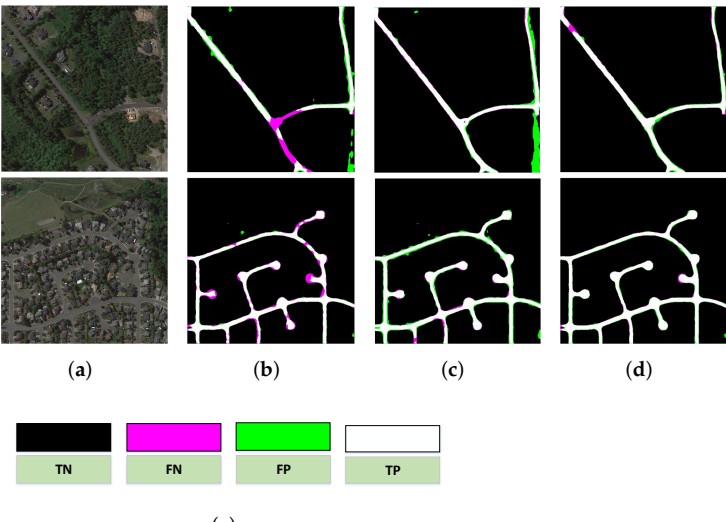

(a)  (b)  (c)  (d)

| TN | FN | FP | TP |

(e)

**Figure 5.** Some qualitative results on dataset 1. (**a**) The original satellite images; (**b**) the results of D-linknet; (**c**) the results of MD-ResUnet; (**d**) the results of our method; (**e**) legend.

The results show SW-GAN achieves better performance in IoU and mAP than the weakly supervised method MD-ResUNet and the fully supervised road extraction D-Linknet, ResUnet, and DeeplabV3. This represents that the proposed SW-GAN can make effective use of the two different kinds of annotations. This model can learn not only the details of a few full annotations but also the potential information of the weak annotations.

From Figures 5–7, we find that SW-GAN can achieve a better performance than the fully supervised method D-linknet and weakly supervised method MD-ResUnet. MD-ResUnet achieves a better performance than the fully supervised method D-linknet in dataset 1 and dataset 2. Additionally, for dataset 3, the fully supervised D-linknet achieves a better performance than that of the weakly supervised method MD-ResUnet. This is because the weakly supervised method MD-ResUnet can extract the distribution and high-order information of the road networks. SW-GAN, which is a semi-weakly supervised method, can achieve both sufficient detail and high-order information of the road networks from both the weak annotations and the full pixel-wise annotations.

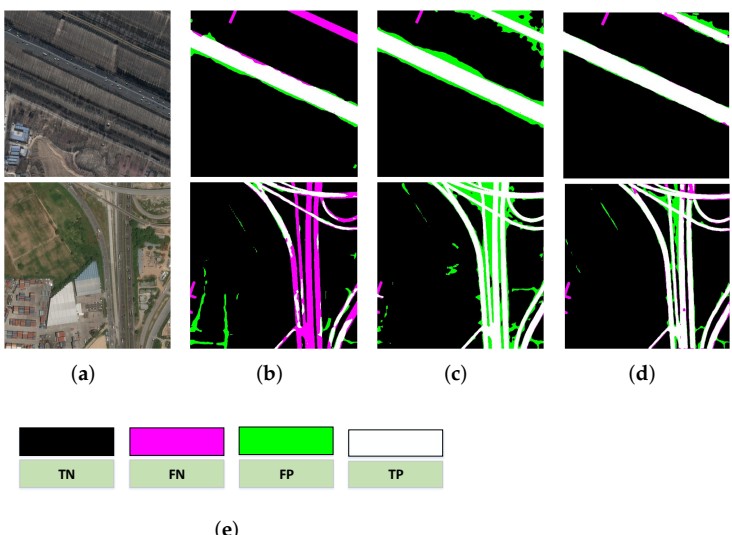

| | | | |
|---|---|---|---|
| TN | FN | FP | TP |

(e)

**Figure 6.** Some qualitative results on Dataset 2. (**a**) The original satellite images; (**b**) The results of D-linknet; (**c**) The results of MD-ResUnet; (**d**) The results of ours. (**e**) legend.

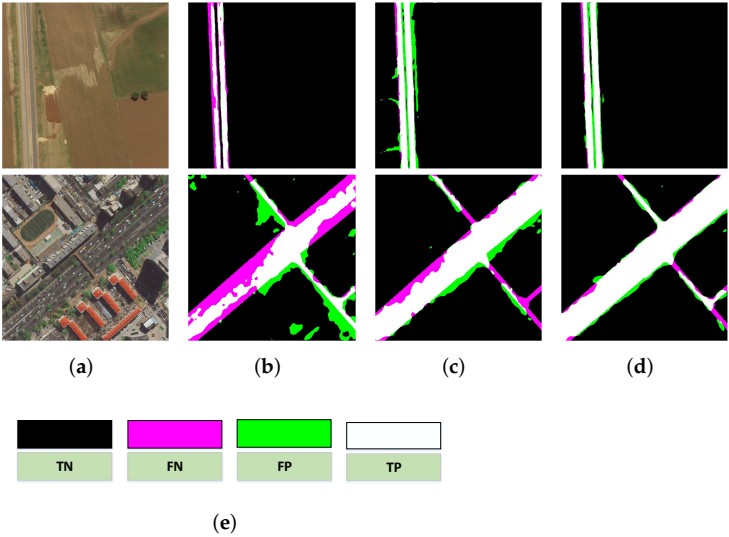

| | | | |
|---|---|---|---|
| TN | FN | FP | TP |

(e)

**Figure 7.** Some qualitative results on Dataset 3. (**a**) The original satellite images; (**b**) the results of D-linknet; (**c**) the results of MD-ResUnet; (**d**) the results of our method; (**e**) legend.

*4.4. Evaluation Using Different Ratios of Full Training Images of SW-GAN*

We conduct experiments using different ratios of pixel-wise annotations and weakly scribbled annotations to evaluate the gap between SW-GAN and the fully supervised model D-linknet on three datasets. When the ratio is 0, the experiment only uses data with weakly labelled annotations to train $G_w + D$. When the ratio is 1, the experiment uses all data with high-quality pixel-wise annotations to train the full supervised network $G_f + D$. The column D-linknet represents the road extraction performance of the fully supervised learning model D-linknet which has been trained with all the high-quality pixel-wise annotation data. The results are shown in Tables 3–5.

**Table 3.** Road Extraction Performance Using Different Ratios of fully supervised Annotation on Dataset 1.

| Ratios | 0.1 | 0.2 | 0.3 | 0.4 | 0.5 | 0.6 | 0.7 | 0.8 | 0.9 | 0 (Weak) | 1 (Full) | 1 (D-Linknet) |
|---|---|---|---|---|---|---|---|---|---|---|---|---|
| IoU | 0.761 | 0.778 | 0.781 | 0.789 | 0.791 | 0.793 | 0.795 | 0.800 | 0.795 | 0.743 | 0.796 | 0.780 |
| mAP | 0.886 | 0.895 | 0.900 | 0.909 | 0.905 | 0.912 | 0.919 | 0.915 | 0.919 | 0.858 | 0.925 | 0.902 |

**Table 4.** Road Extraction Performance Using Different Ratios of fully supervised Annotation on Dataset 2.

| Ratios | 0.1 | 0.2 | 0.3 | 0.4 | 0.5 | 0.6 | 0.7 | 0.8 | 0.9 | 0 (Weak) | 1 (Full) | 1 (D-Linknet) |
|---|---|---|---|---|---|---|---|---|---|---|---|---|
| IoU | 0.618 | 0.637 | 0.658 | 0.669 | 0.664 | 0.663 | 0.6694 | 0.665 | 0.671 | 0.586 | 0.670 | 0.652 |
| mAP | 0.718 | 0.738 | 0.748 | 0.745 | 0.757 | 0.767 | 0.770 | 0.772 | 0.771 | 0.731 | 0.779 | 0.751 |

**Table 5.** Road Extraction Performance Using Different Ratios of fully supervised Annotation on Dataset 3.

| Ratio | 0.02 | 0.04 | 0.08 | 0.1 | 0.2 | 0.4 | 0.6 | 0.8 | 0.9 | 0 (Weak) | 1 (Full) | 1 (D-Linknet) |
|---|---|---|---|---|---|---|---|---|---|---|---|---|
| IoU | 0.677 | 0.687 | 0.698 | 0.701 | 0.714 | 0.716 | 0.711 | 0.715 | 0.716 | 0.656 | 0.719 | 0.710 |
| mAP | 0.737 | 0.779 | 0.795 | 0.804 | 0.812 | 0.813 | 0.806 | 0.804 | 0.812 | 0.705 | 0.806 | 0.808 |

The results from Tables 3–5 show that the IoU and mAP of SW-GAN can achieve close performance to the fully supervised method D-linknet just using 30% full annotations. IoU and mAP increase rapidly with the increase of the pixel-wise annotations from 0 to 30%. Then, IoU and the mAP stayed unchanged or slightly increased with the increase of annotations from 30% to 100%. These results prove that adding a few full annotations into SW-GAN can rapidly improve the performance of the road extraction. Additionally, by using a small set of the full annotation data, the proposed model can gain almost the same results as a fully supervised model using all full pixel-wise annotations. This is because the weakly supervised generator can learn the distribution and high-order information from weakly supervised annotation data effectively. Furthermore, the fully supervised generator can learn the details of the road network in a fully supervised learning fashion. From the experimental results, we can conclude that the semi-weakly supervised method is of great importance for road extraction from remote sensing imagery, since it can effectively learn potential information from both a large weakly supervised dataset and a small fully supervised dataset.

*4.5. Ablation Study*

To verify the rationality of the proposed network, we conduct ablation experiments in this section. We choose Dataset 1 to do the ablation experiments with 20% full pixel-wise annotations and 80% weakly centerline annotations. The experimental results are shown in Table 6.

**Table 6.** Ablation experiments on Dataset 1 (bold: best).

| Model | $G_f$ | $G_f+G_w$ | $G_f+D$ | SW-GAN |
|---|---|---|---|---|
| IoU | 0.7293 | 0.7627 | 0.7387 | **0.7781** |
| mAP | 0.8362 | 0.8851 | 0.8395 | **0.8951** |

Table 6 shows that the network using only the fully supervised generator $G_f$ achieves the lowest performance in IoU and mAP compared to others. When adding the weakly supervised generator $G_w$ or the discriminator $D$ to $G_f$, IoU and mAP will increase. That is

because the weakly supervised generator can extract potential information from weakly annotated data and the discriminator can enhance the performance of the fully supervised generator by adversarial learning process. The proposed model SW-GAN, which consists of $G_f$, $G_w$ and $D$, obtains the best performance.

### 4.6. The Influence of the Hyper Parameters to the Semi-Weakly Supervised Road Extraction

To find proper value for hyper parameter $\lambda_1$ and $\lambda_2$ of SW-GAN, we test different $\lambda_1$ and $\lambda_2$ to evaluate the extraction performance. The candidate values of $\lambda_1$ are set as 0.0001, 0.01, 0.1, 1, 10 and 100; and the candidate values of $\lambda_2$ are set as 0.02, 0.1, 0.4, 2, 10, 50, 250 and 1000. We combine the different values of the two parameters and obtain 48 different groups of model hyper parameters. On the one hand, deep learning model experiments are very time-consuming. On the other hand, dataset 1 is a widely used typical road extraction data set. The hyper parameters obtained by experiments on dataset 1 can also be regarded as typical, which can be applied to road extraction data sets with similar statistical characteristics. Therefore, we test 48 groups of hyper parameters one-by-one only on dataset 1 and utilize 20% pixel-wise annotations and 80% weak centerline annotations. The results are shown in Table 7.

**Table 7.** The IoU performance of the proposed model with different hyper parameters (bold: best).

| $\lambda_1$ \ $\lambda_2$ | 0.02 | 0.1 | 0.4 | 2 | 10 | 50 | 250 | 1000 |
|---|---|---|---|---|---|---|---|---|
| 0.0001 | 0.735 | 0.747 | 0.766 | 0.769 | 0.770 | 0.763 | 0.752 | 0.741 |
| 0.001 | 0.749 | 0.751 | 0.771 | 0.774 | 0.772 | 0.769 | 0.761 | 0.743 |
| 0.1 | 0.750 | 0.758 | 0.775 | **0.778** | 0.774 | 0.770 | 0.760 | 0.743 |
| 1 | 0.737 | 0.742 | 0.767 | 0.766 | 0.763 | 0.764 | 0.755 | 0.741 |
| 10 | 0.721 | 0.724 | 0.731 | 0.738 | 0.728 | 0.732 | 0.713 | 0.710 |
| 100 | 0.454 | 0.466 | 0.463 | 0.464 | 0.461 | 0.462 | 0.452 | 0.474 |

From Table 7 we can see that when $\lambda_1$ is set as 0.1 and $\lambda_1$ is set as 2, the proposed model achieves the best performance. If the value of $\lambda_1$ and $\lambda_2$ is too large or too small, the performance of the proposed model degrades. The reason for this is that when $\lambda_1$ is too large, the generator of SW-GAN is only supervised by the fully supervised data set. While the $\lambda_1$ is too small, the training process of SW-GAN cannot obtain a strong supervised signal from the labeled data. Similarly, if $\lambda_2$ becomes large enough, the results are closer to the road extraction method using only the weakly supervised network. Additionally, when $\lambda_2$ is too small, the results of SW-GAN are almost equal to a typical GAN that only uses the fully supervised generator $G_f$ and the discriminator $D$.

From the experiments and analysis of Sections 4.3–4.5, we can conclude that:

1. The proposed SW-GAN outperforms both the state-of-the-art fully supervised methods ResUNet and D-Linknet as well as the weakly supervised method MD-ResUnet in road extraction.

2. Compared with those methods, supervised by large amounts of pixel-wise full annotations, the proposed SW-GAN can achieve very close performance using only small amounts of pixel-wise full annotations and large amounts of weak centerline annotations.

3. The more pixel-wise annotations fed into SW-GAN, the better the road extraction performance will be. However, the performance increases more and more slowly with the increase of the full annotation data.

## 5. Conclusions

In this paper, a semi-weakly supervised framework is proposed to extract road networks from remote sensing imagery, which uses large amounts of weak annotation data and only a small number of full pixel-wise annotation data. The performance of the proposed method is evaluated on three datasets in various settings both quantitatively and qualitatively. The method achieves very close performance compared to the state-of-the-art

methods using full annotations. In future, to obtain more practical road networks for our daily life, we would like to focus on road extraction using 3-D data.

**Author Contributions:** H.C., C.D. and S.P. proposed the network architecture design, S.P. and S.W. performed the experiments and analysed the results. H.C. and S.W. wrote the paper. C.D. and J.L. revised the paper and provided valuable advises. All authors have read and agreed to the published version of the manuscript.

**Funding:** This work is supported in part by the National NSF of China under grants No. U19A2058, No. 41971362, No. 41871248 and No. 62106276.

**Acknowledgments:** The authors would also like to thank the anonymous referees for their valuable comments and helpful suggestions.

**Conflicts of Interest:** The authors declare no conflict of interest.

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
