# Peer review of "SW-GAN: Road Extraction from Remote Sensing Imagery Using Semi-Weakly Supervised Adversarial Learning"

_remotesensing, doi:10.3390/rs14174145_

Round 1

Reviewer 1 Report

The authors present a novel method to use a weak annotated large dataset from a public source in combination with a small completely annotated dataset. The goal is to achieve a performance close to a fully annotated dataset. 

The topic is well motivated and well described. The approach is from my point of view creative and interesting from a scientific point of view. 

The title of the article has created the expectation that the paper is working in the domain of autonomous driving which is not the case. I would therefore recommend to mention that the paper is based on aerial and not ego-centric data.

Overall the paper needs some minor corrections but should be considered for publication.

In Table 1 the annotation type should be manual. 

Several references are incomplete. Reference 8 does not mention the conference. Reference 18 does not include the complete conference description. Same for 28 and 29.

Author Response

Dear Editors and Reviewers:

Thank you for your letter and the reviewers’ comments concerning our manuscript entitled “SW-GAN: Road Extraction Using Semi-weakly-supervised Adversarial Learning” (ID: remotesensing-1844296). Those insightful comments are valuable for revising and improving our paper. We have proofread comments carefully and tried our best to address the concerns. Revised portion are marked in red in the paper.
Major corrections in the paper and the response to the reviewer’s comments are as follows:
Reviewer #1: We thank the reviewer for the careful review and helpful comments.
Q1.    The title of the article has created the expectation that the paper is working in the domain of autonomous driving which is not the case. I would therefore recommend to mention that the paper is based on aerial and not ego-centric data.
A1.    Considering the Reviewer’s suggestion, we have changed the title of our manuscript to “SW-GAN: Road Extraction from Remote Sensing Imagery Using Semi-weakly-supervised Adversarial Learning”.
Q2.    In Table 1 the annotation type should be “manual”.
A2.    We apologize for any confusion typos may have caused. We have revised the manuscript accordingly.
Q3.    Several references are incomplete. Reference 8 does not mention the conference. Reference 18 does not include the complete conference description. Same for 28 and 29.
A3.    We have checked and completed the information of all the references according to the Reviewer’s comments.

Reviewer #2: We thank the reviewer for the careful review and helpful comments.
Q1.    I think that Equation (1) should be commented on in the text. The same for Equations (3) and (4).
A1.    We have added some descriptions about Equation (1), (3) and (4) in the manuscript according to the Reviewer’s comments.
Q2.    Table 1 should be enriched with sizes of train and test sets after the augmentation process.
A2.     We have added the sizes of train and test sets after the augmentation according to the Reviewer’s comments.
Q3.    Why did you use SGD instead of the more popular and modern ADAM optimizer?
A3.    In fact. We used SGD, Adagrad, RMSprop and ADAM as optimizer to test our model, and found the performance of the proposed model was almost equal. And the validation loss of our model was a little lower than the other optimizers when SGD was used. Therefore, we adopted SGD as the optimizer.
Q4.    Could you add more information on how did you select hyper-parameters?
A4.    We apologize for not explaining how to select hyperparameters clearly, the details are stated in subsection 4.6 instead of subsection 4.4. We have revised subsection 4.6 to state our method clearly. We found when lambda_1 is 0.1 and lambda_2 is 2, the performance of our model is the best, so we chose them as the value of model hyperparameters for evaluation and comparison with the state-of-the-art methods.
Q5.    Section 4.6 is not confirmed. Experiments only on one data set are not enough to confirm the authors' hypothesis.
A5.    In fact, for most deep learning based models, the best hyperparameters may be different for different data set. The best method may be to determine the most appropriate hyperparameters for each data set, yet it cannot be applicable in practice. For our work, on the one hand, deep learning model experiments are very time-consuming, the workload of 48 groups of experiments on each of the three different datasets is very large. on the other hand, Dataset 1 is a wildly-used typical road extraction data set. The hyperparameters obtained by experiments on Dataset 1 can also be regarded as typical, which can be extended to road extraction data sets with similar statistical characteristics. Therefore, we only test 48 groups of hyperparameters one by one only on Dataset 1 to determine the best group of hyperparameters. We test our model on Dataset 1, 2 and 3 with the same hyperparameters, and the performance of our model outperforms the state-of-the-art methods. And we can conclude that the value of the hyperparameters is within a reasonable range for the proposed model, although it may be not the best ones.
Q6.    The authors did not indicate the code repository, therefore replicating the results seems difficult.
A6.    We will organize and add some necessary comments to our source code. After that, we will open it at https://github.com.
Q7.    A lot of typos need correction. In general, I highly recommend professional proofreading.  
A7.    We apologize for the typos in the manuscript. We have carefully checked the manuscript and corrected the typos. After that, we also have asked someone to do proofreading for the manuscript.
Q8.    Authors mention Fig. 2 on page 7 but Fig 2 appears on page 9. In my opinion, this figure should be earlier; At the end of most equations should be a comma or full stop; We should enumerate only equations which we cite in the text.
A8.    Considering the Reviewer’s suggestion, we have revised the manuscript.
Q9.    In my opinion Figure 4 is not sufficiently commented on in the text.
A9.    We have added more detail to describe Figure 4, according to the Reviewer’s comments.
Q10.    I propose to add the definition of IoU and mAP.  
A10.    We have added the definition of IoU and mAP to the manuscript according to the Reviewer’s comments.

We tried our best to improve the manuscript and made changes in the manuscript. These changes will not influence the content and framework of the paper. And here we did not list all the changes but marked the corresponding parts in red in the revised paper.
We appreciate for Editors/Reviewers’ careful review and hope for your favorable consideration regarding this manuscript. 

Best regards, 
Hao Chen,  
Shuang Peng, 
Chun Du,
Jun Li
and Songbing Wu

Reviewer 2 Report

Major remarks:

- I think that Equation (1) should be commented on in the text. The same for Equations (3) and (4).
- Table 1 should be enriched with sizes of train and test sets after the augmentation process.
- Why did you use SGD instead of the more popular and modern ADAM optimizer?
- Could you add more information on how did you select hyper-parameters (page 15, line 290)?
- Section 4.6 is not confirmed. Experiments only on one data set are not enough to confirm the authors' hypothesis.
- The authors did not indicate the code repository, therefore replicating the results seems difficult.

Minor remarks:
- A lot of o typos need correction. In general, I highly recommend professional proofreading.
- Authors mention Fig. 2 on page 7 but Fig 2 appears on page 9. In my opinion, this figure should be earlier.
- At the end of most equations should be a comma or full stop.
- We should enumerate only equations which we cite in the text.
- In my opinion Figure 4 is not sufficiently commented on in the text.
- I propose to add the definition of IoU and mAP.

Author Response

(The authors gave the same response as above.)
